# Numerical Study on the Dependency of Microstructure Morphologies of Pulsed Laser Deposited TiN Thin Films and the Strain Heterogeneities during Mechanical Testing

**DOI:** 10.3390/ma14071705

**Published:** 2021-03-30

**Authors:** Konrad Perzynski, Grzegorz Cios, Grzegorz Szwachta, Piotr Bała, Lukasz Madej

**Affiliations:** 1Faculty of Metals Engineering and Industrial Computer Science, AGH University of Science and Technology, Mickiewicza 30 Ave., 30-059 Krakow, Poland; pbala@agh.edu.pl (P.B.); lmadej@agh.edu.pl (L.M.); 2Academic Centre for Materials and Nanotechnology, AGH University of Science and Technology, Mickiewicza 30 Ave., 30-059 Krakow, Poland; grzegorz.cios@agh.edu.pl (G.C.); gjs@agh.edu.pl (G.S.)

**Keywords:** pulsed laser deposition, thin films, digital material representation, kinetic Monte Carlo

## Abstract

Numerical study of the influence of pulsed laser deposited TiN thin films’ microstructure morphologies on strain heterogeneities during loading was the goal of this research. The investigation was based on the digital material representation (DMR) concept applied to replicate an investigated thin film’s microstructure morphology. The physically based pulsed laser deposited model was implemented to recreate characteristic features of a thin film microstructure. The kinetic Monte Carlo (kMC) approach was the basis of the model in the first part of the work. The developed kMC algorithm was used to generate thin film’s three-dimensional representation with its columnar morphology. Such a digital model was then validated with the experimental data from metallographic analysis of laboratory deposited TiN(100)/Si. In the second part of the research, the kMC generated DMR model of thin film was incorporated into the finite element (FE) simulation. The 3D film’s morphology was discretized with conforming finite element mesh, and then incorporated as a microscale model into the macroscale finite element simulation of nanoindentation test. Such a multiscale model was finally used to evaluate the development of local deformation heterogeneities associated with the underlying microstructure morphology. In this part, the capabilities of the proposed approach were clearly highlighted.

## 1. Introduction

The deposition process of thin films by laser ablation has been known since the 1970s [1]. This method is used in a wide range of industrial applications: from superconductor production, through the manufacturing of forming tools coatings, up to the biocompatible materials for medical implants [2,3].

Deposition processes are especially interesting in the latter applications as they can provide thin films at the medical equipment and increase both its strength and bio-protection properties [2]. However, in these applications, development of any internal defects in the thin film due to, e.g., local strain localization occurring during exploitation conditions, is unacceptable. Such defects can cause a build-up of biological material and deterioration of strength properties, which may be hazardous for a patient. That is why morphology of deposited thin layers must ensure all mechanical expectations required for their applications. Presently, there is a wide variety of methods that can be used for deposition purposes. The primary representative of these technologies is the pulsed laser deposition (PLD) method [4], which is a modification of the standard physical vapor deposition (PVD) approach. The main idea of PLD is based on evaporation and ionization of the surface atoms by a high-power laser beam that is periodically focused on a target. Particles are first detached by a laser and then strike with high speed into the surface of the substrate materials and start to nucleate and grow. This process provides possibilities to obtain layers at different kinds of engineering materials. The stoichiometry of the target material is reflected in the films very well, improving adhesion between the layer and the substrate. These properties result mainly from the fact that particles in an atomic beam have considerable kinetic energies (0.1–100 eV), which increase the diffusion rate of adatoms at the surface [5]. PLD can be divided into three stages: ablation, transportation of atoms through a chamber, and deposition onto a substrate surface. Experimentally, each pulse lasts for a few nanoseconds, and the time between two pulses is of the order of a second. That is why the thin layers’ growth mechanism by a pulsed laser deposition is an extremely complex process. For this reason, the development of a specific deposition technology for a given material, often involves long-term research aimed at an empirical determination of the required process parameters. 

That is why, at first, authors decided to develop a numerical model of the PLD deposition process, which can support experimental research and also provide reliable data on thin film morphologies for further studies of their behavior under exploitation conditions. Commonly used numerical models of deformation neglect the inner structure of deposited thin films [6,7], and is what depreciates the quality of obtained data. Deposited material under, e.g., loading conditions, is usually defined as isotropic, without taking into account columns and surface wrinkling, which are commonly observed under structural investigation. Thus, simplified models cannot give sufficient information about material resistance to deformation. Therefore, the development of the model, which precisely maps thin films’ morphologies and inner structures during modelling of exploitation conditions, seems extremely relevant and important, to obtain results comparable with those from experimental investigations.

Reliable digital material representation (DMR) of the thin film microstructure morphology numerical model of the deposition process was developed first. It provides a digital representation model [8,9] of layers for subsequent numerical simulations of deformation under nanoindentation conditions [10] as a function of deposition process parameters. As a result, the complex nanoindentation model will provide a basic understanding of local heterogeneous material response to deformation conditions. 

## 2. Numerical Modelling of the PLD Process

The first 2D numerical models of deposition aimed at capturing underlying physics can be found in the literature from early 1970s [11]. Presently, two major types of approach can be distinguished: the deterministic, and the stochastic ones. The most common examples of deterministic models are based on the molecular dynamics method [12,13,14]. This method describes the movement of individual atoms or material particles by the Newton equation of motion. However, to calculate forces and energies between atoms, a set of interatomic potentials must be defined. The molecular dynamics (MD) method can directly address mechanisms of deposition at the nanoscale. However, due to a small length scale and available computational resources, it only allows for investigating a very local material behavior, far beyond the industrial expectations. That is why stochastic models of deposition are more frequently used during an investigation. 

The first group of stochastic methods is based on the cellular automata (CA) technique. The CA technique’s main idea is to divide a specific part of the material into a one-, two-, or three-dimensional lattice of finite cells. Each cell in the CA space is also surrounded by neighbors, which affect one another. The cells interactions within the CA space are based on the knowledge defined while studying a particular phenomenon. An example of the CA’s epitaxially thin layers growth simulation was presented in authors’ earlier works [15,16,17,18,19]. These approaches can be classified as Random Deposition Models (RDM) [20,21], and consist of three main steps: random deposition of particles at the growing surface, calculation of the total energy of each particle with the migration of mobile particles along the surface, and eventually, desorption of particles from the surface.

The second group is based on the Monte Carlo (MC) method. Models, which belong to that group provide a possibility to describe the evolution of complex systems, with some simplifications, during calculations [22,23,24,25]. They are based on a random sampling of the examined quantity with the use of its analytical distributions. However, the MC method does not directly take into account the elapse of time, which is particularly important in the description of systems in which events occur concurrently and are mutually dependent. Because a thin film growth belongs to such systems, therefore, a modified version of the MC method was developed to take into account the process kinetics, and is called the kinetic Monte Carlo (kMC).

A group of the kMC methods describes the progression of complex systems by the identification of all possible events and assigning to them the probability of occurrence. For that, the method requires knowledge about rates of each event, which are determined based on values of energy barriers that the system has to overcome to move to a new configuration. Statistically, the most likely events are selected more often. The great advantage of the kMC method is consideration of the physical time of the process during a simulation. With that, it is possible to take into account many competing mechanisms occurring during a deposition process: atoms deposition, diffusion on the surface, island formation, connection/disconnection to/from existing atoms’ islands, ascending/descending on/from the existing atoms’ islands and evaporation.

Therefore, the kMC approach was selected for the current investigation focused on the implementation of the PLD deposition model of TiN thin films.

### Formulation of the kMC PLD Deposition Model

To describe the evolution of the growing layer during the PLD, mentioned earlier, important processes were grouped into two elementary phenomena:

Adsorption—particles from plasma flux are attached to the substrate surface due to the weak van der Waals forces; 

Surface-diffusion—particles at the surface can change their sites in favor of energy minimization; 

A concept of all mentioned elementary events, which are considered during the kMC model development, is shown in Figure 1.

As mentioned, the key information required by the kMC algorithm is related to the rates of these fundamental events, which can be described by the following set of equations:

Adsorption rate:(1)$$r_{ads}=\frac{D}{a}$$
where *D*—deposition rate, *a*—dimension of an elementary cell.

Surface diffusion rate is given by an Arrhenius-type expression:(2)$$r_{diff}=\;\alpha Te^{\frac{-\Delta E}{kT}}$$
where k—Boltzmann constant, T—relative substrate temperature, α—adatom vibration frequency.

Change of an occupied site is driven by an energy difference (activation energy): (3)$$\Delta E=\;E_{1}-E_{0}$$
where $E_{1}$—energy of a particle after a hypothetic change, $E_{0}$—current energy of a particle.

The energy of a given particle is defined as:(4)$$E=\underset{i}{\sum }E_{i}$$
where $E_{i}$—binding energy between a considered particle and a particle from a neighborhood, which is placed at a site i.

The schematic block diagram of the implemented kMC algorithm is presented in Figure 2.

Following the kinetic Monte Carlo method for each calculation step, a list of possible events in the system is computed along with probabilities of their occurrence. Then, based on a random number from 〈0,R), where R is an accumulated probability of all events, a single event is selected and applied to the system. 

Therefore, the kMC consists of several steps:(1)Creation of a list of all possible events in the system and calculation of a likelihood of their occurrence $r_{i}$.(2)Calculation of the sum of probabilities of all events $R=\;\overset{i}{\underset{j=0}{\sum }}r_{j}$.(3)Random selection of a number in a range 〈0,R).(4)Each event is placed on a stack. Graphically (Figure 3), the height of a particular event represents its probability of occurrence. An overall height stack is thus equal to a cumulated probability of all considered events—R. A randomly chosen number u unambiguously indicates the event, which will be applied to the system. Selection of the event is shown in Figure 3. (5)Transposition of the system to a new state by applying the selected event.(6)Updating the time counter by Δt=1/R.

From the implementation point of view, an aspect that requires particular attention is optimising the algorithm towards reducing computing time. In the classical kMC approach, a list of all events is recreated at each time step, which is a severely CPU time-consuming task. To improve the algorithm’s performance, a list of events can be initialized only once and later, depending on an event which is selected accordingly, updated by: Adding events, which become possible;Removing obsolete events;Updating probabilities of all events, which could be affected by a previous change in the system.

Keeping track of all events, which will be affected by applying a new event to the system, is challenging. The simplest way to achieve this is to recalculate the probability of directly related events to the extent of the existing neighborhood. This is because some types of events, i.e., surface diffusion, depend not only on the particles’ actual configuration, but also on the configuration after a hypothetical movement of the particle. An example of this procedure is shown in Figure 4. The considered neighborhood has a radius of a single particle. The red border represents the range of a doubled neighborhood. Affected particles are thus located in the doubled neighborhood before and after the particle movement.

## 3. Kinetic Monte Carlo Simulations of the PLD Process 

The developed kMC deposition model was used in the work to simulate the growth of the TiN thin film on the single-crystal Si substrate. Deposition process parameters, namely, substrate temperature $T_{sub}=200\;{}^{\circ}C$ and deposition rate 0.05 nm/s, were selected to obtain columnar growth. Model parameters presented in Table 1 were selected based on the literature findings [26,27,28] and a series of initial simulations. 

The TiN thin film with the dimension of 90 nm × 90 nm × 90 nm was obtained from the kMC simulation. Example of the DMR model of thin film during kMC simulation is shown in Figure 5.

As presented in Figure 6, the shape of generated columns in the thin films is highly irregular. Bottom columns are characterized by a lower surface area than the columns in the thin film’s upper part. To show a change in the columns’ geometry and the height of the final sample, a surface area of each column at subsequent cross-sections (22, 45, and 67 nm) was calculated and presented in Figure 6.

Additionally, in Figure 7, it can be seen that the average height of the four highest columns is between 87 to 91 nm. The average width of those columns in the middle is between 18 to 25 nm, but in the top part, an average width increases and is between 19 to 28 nm (Figure 6). All columns have a V shape, which can also be observed through microscopic investigations.

## 4. Experimental Investigation

To validate the PLD model predictions, a 90 nm TiN thin film was deposited in the laboratory conditions on the Si(100) substrate using the 248 nm excimer laser system (Coherent COMPexPro 110F, Santa Clara, CA, USA) operated at an energy density of ~3 J·cm^−2^, a pulse width of 20 ns, and a repetition rate of 10 Hz. The target was a disc with 2.54 cm in the diameter and 0.5 cm in the thickness. The initial pressure in the chamber was set to 5 × 10^−7^ Torr. The silicon substrate was subjected to an ultrasonic cleaning procedure for 10 min in acetone and 10 min in methanol and finally, etched for 5 min in 10% HF. The substrate was placed parallel to the target material surface at a distance of 5 cm. The deposition temperature and nitrogen partial pressure were 200 °C and 1 × 10^−5^ Torr, respectively [9,29]. Process settings closely replicated the conditions selected during numerical modelling presented earlier. 

Investigation of the TiN thin layer morphology was carried out by the transmission electron microscopy (TEM, Tecnai TF 20 X-TWIN, FEI, Hillsboro, OR, USA). The sample was prepared for this investigation by a focused ion beam (FIB) technique (Quanta 3D 200i, FEI). The FIB preparation included an electron beam Pt deposition at the beginning and a low accelerating voltage cleaning as the final step. To evaluate deposited columns’ thickness, a set of dark field images was taken within the same area using a different peak lying on the first ring of the diffraction pattern. Observed columns with marked investigated locations and corresponding dimensions are presented in Figure 8 and Table 2, respectively.

As seen in Figure 7 and Figure 8 and Table 2, the numerical model predictions appropriately replicate experimental observations. Therefore, the developed PLD model can be used to generate realistic morphologies of thin films for further numerical investigations of their behavior under, e.g., loading conditions based on the mentioned digital material representation concept. 

## 5. Numerical Nanoindentation Test Based on the Explicit Representation of Thin Films Morphologies

The nanoindentation test, which is commonly used to evaluate thin films’ mechanical properties, was selected as a case study for the present investigation. The partially-coupled concurrent multiscale methodology was used due to a significant length scale difference between the microscopic model of the nanoindentation test, and nanoscale model based on the digital material representation. In this approach, a particular area of interest from the macroscale model is selected and recalculated with a refined mesh to obtain a more detailed solution. The microscale model contains information on the sample geometry and boundary conditions of the nanoindentation test. Additional features related to the nanoscale model, e.g., columnar morphology, are initially excluded at this length scale. On the other hand, the nanoscale model is an arbitrary cutout taken from the micromodel, and it takes into account the digital model of the thin film morphology obtained from the developed kMC algorithm. In this procedure, the microscale model is first calculated within the finite element (FE) ABAQUS/Explicit code. The refined nanoscale model is resubmitted for simulation with displacement boundary conditions taken from the micro model simulation. A schematic description of this multiscale technique, applied for the nanoindentation test, is presented in Figure 9.

During the simulation, a diamond nanoindenter is assumed to be a deformable body with elastic modulus and Poisson’s ratio taken from the literature [30]. The micromodel of the thin film was discretized with 150,000 8-node tetrahedron elements. Additionally, a displacement of the Si(100) substrate was fixed by the rigid tool situated at the bottom of the sample.

Examples of results obtained from the micromodel are shown in Figure 10. As mentioned, they are then used as boundary conditions for the nanoscale model, which include a columnar morphology of the thin film.

As mentioned, the nanoscale model is based on the thin film morphology presented in Figure 7. The possibility of assigning material properties to particular TiN columns was described in the previous work [29]. The generated morphology was also subjected to a discretization algorithm. The non-uniform mesh was created using a *DMRmesh* software (v0.9) [31]. The FE mesh (Figure 11) is highly refined along the columns’ boundaries to adequately capture solution gradients that are expected in these regions due to differences in the hardening behavior of subsequent columns. 

The thin film DMR model was discretized with 631,000 four-node linear tetrahedron (C3D4) elements (Figure 11). Such nanoscale model was then located in 9 different location within the deformation area, according to Figure 9. Examples of results presenting the influence of a columnar morphology on inhomogeneities in both stress and strain fields that may result in, e.g., local fracture, are presented in Figure 12 and Figure 13.

In the TiN film, especially between columns, large strain and stress irregularities can be observed at larger nanoindenter displacements. Prediction of these regions is important, as stress and strain concentration zones can easily change into fracture initiation zones, and lead to a destruction of the thin film that can be observed experimentally in Figure 14.

## 6. Discussion

The V shape columnar TiN structure investigated within the work (Figure 8) is often reported in the literature when a low Si substrate temperature is considered [32,33,34]. As presented, such a columnar structure of a thin film can be quite heterogeneous. Therefore, precise control and understanding of a deposition operation are required to obtain desired thin film morphology. The developed kMC deposition model can serve as a support tool for such an investigation as it provides a reliable and explicit representation of columns. It can be used as a tool for the preliminary assessment of applied process parameters to evaluate their influence on the final thin film morphology. Additionally, such digital model can be applied to more complex analysis of film behavior under further processing conditions, by means of the FE numerical simulations. An example of a numerical analysis of the TiN thin film from the mechanical properties point of view during the nanoindentation test was presented within the paper as a case study. Such numerical investigation can complement experimental investigations reported, e.g., in [35,36,37]. Since the column boundaries have been identified as the fracture initiation zones, the fracture resistance of the inter-column boundaries can be considered as the fracture resistance of the entire thin film. With the presented concept of combining the kMC deposition model and digital material representation finite element simulations, it is possible to extend research in this area.

## 7. Conclusions

Based on the presented results, it can be concluded that:The kinetic Monte Carlo method is an adequate and feasible technique for numerical simulation of the PLD process and provides a reliable digital representation of microstructure morphology;The presented kMC PLD model can be adjusted to design the deposition processes of different nanolayered structures;The digital material representation model of the deposited thin films allows predicting of inhomogeneities in stress/strain fields under deformation conditions;Predicted local heterogeneities, especially in the interface area and along columns boundaries, can be further used to study fracture initiation and propagation.

## Figures and Tables

**Figure 1 materials-14-01705-f001:**
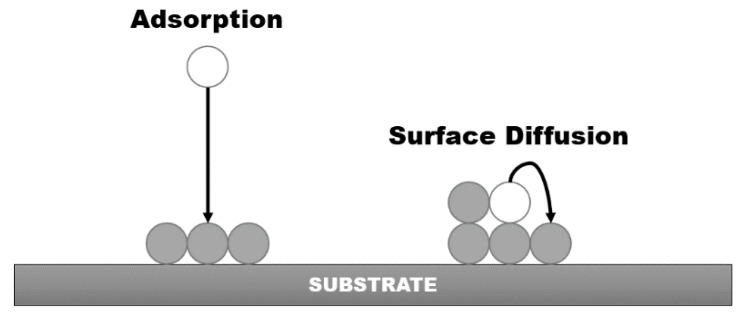
Elementary events during the PLD accounted for by the developed kMC model.

**Figure 2 materials-14-01705-f002:**
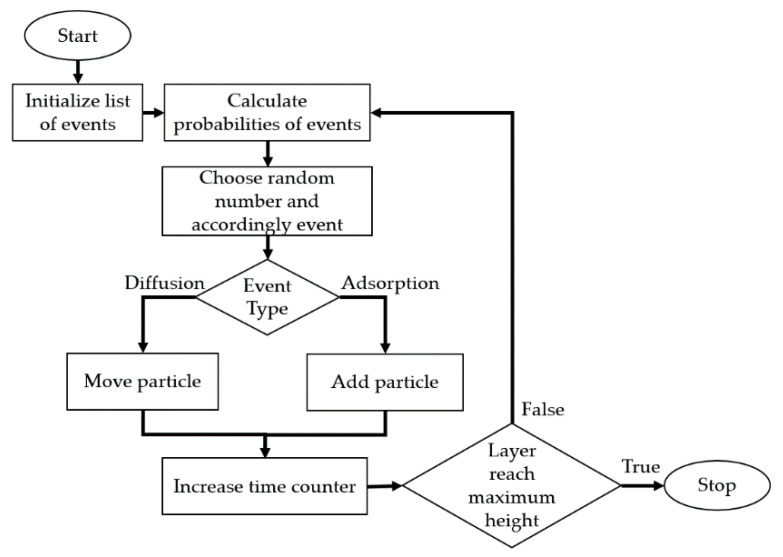
Block diagram showing the kMC algorithm of the PLD.

**Figure 3 materials-14-01705-f003:**
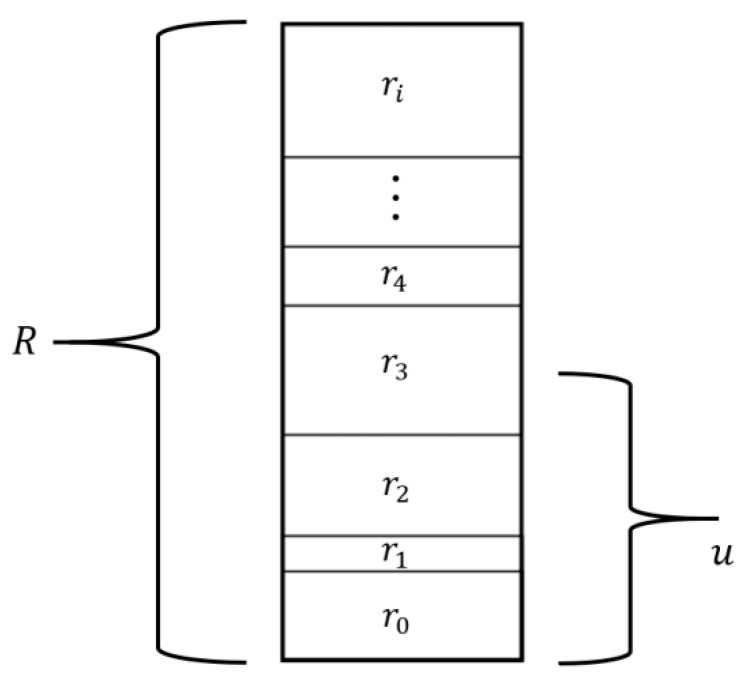
Method of selecting an event from the available list in the kMC algorithm.

**Figure 4 materials-14-01705-f004:**
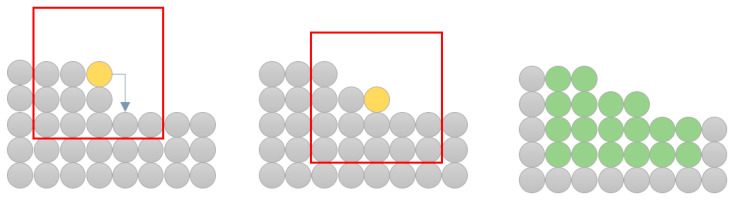
Procedure of choosing particles, and which interrelated events probabilities could be affected (marked as green) after applying a surface diffusion to the system (marked as yellow).

**Figure 5 materials-14-01705-f005:**
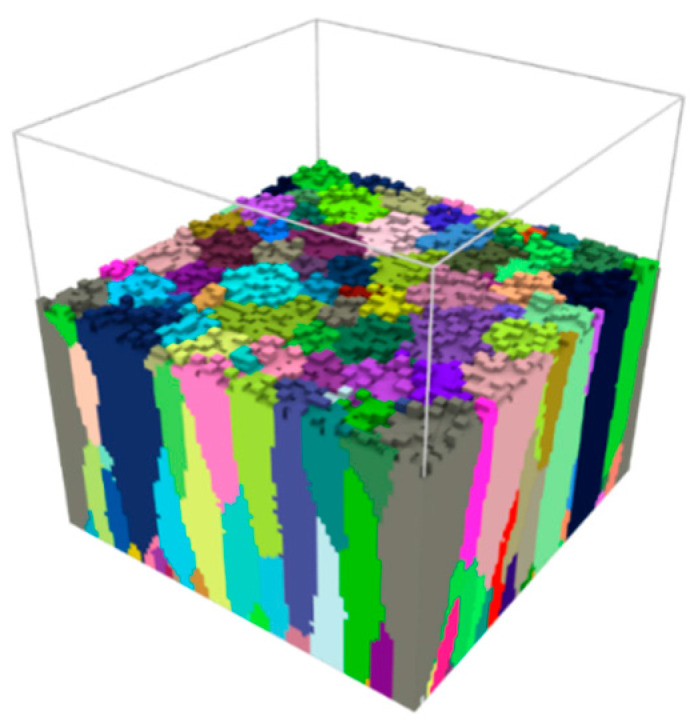
Microstructure morphology during the kMC simulation of the TiN thin film deposition.

**Figure 6 materials-14-01705-f006:**
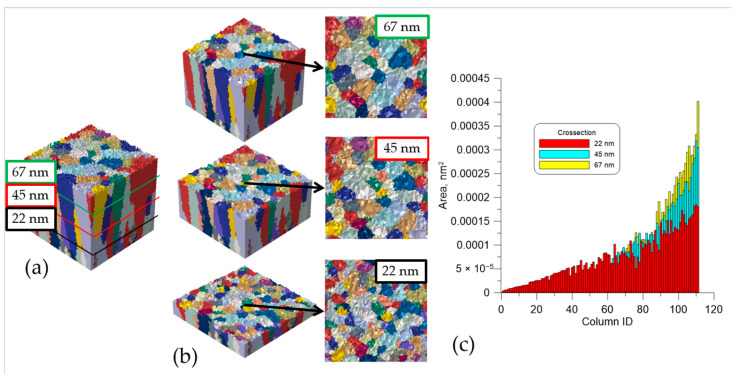
(**a**) position of the cross-section in the final DMR sample, (**b**) illustration of columns at particular cross-sections, and (**c**) diagram representing areas of subsequent columns at particular cross-sections.

**Figure 7 materials-14-01705-f007:**
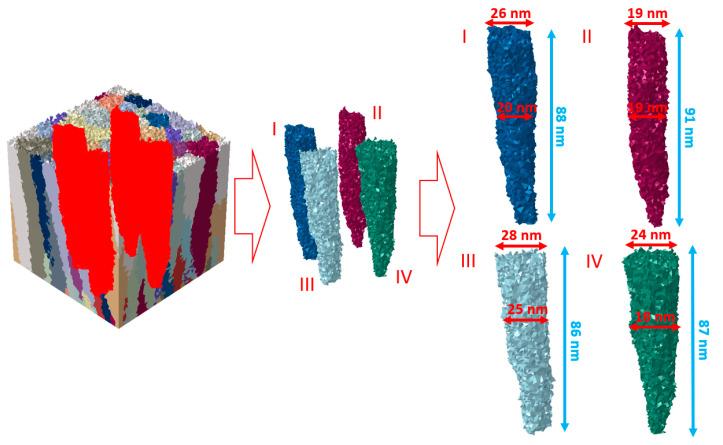
Dimensions of the columns obtained from PLD simulation.

**Figure 8 materials-14-01705-f008:**
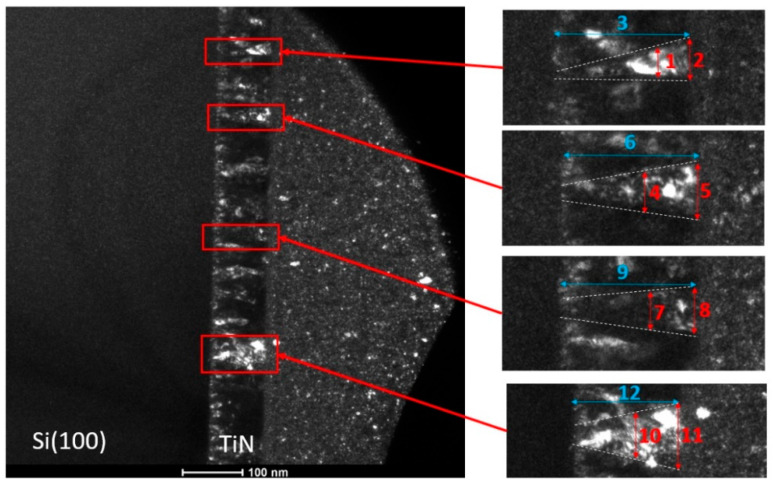
TEM images with marked investigated locations.

**Figure 9 materials-14-01705-f009:**
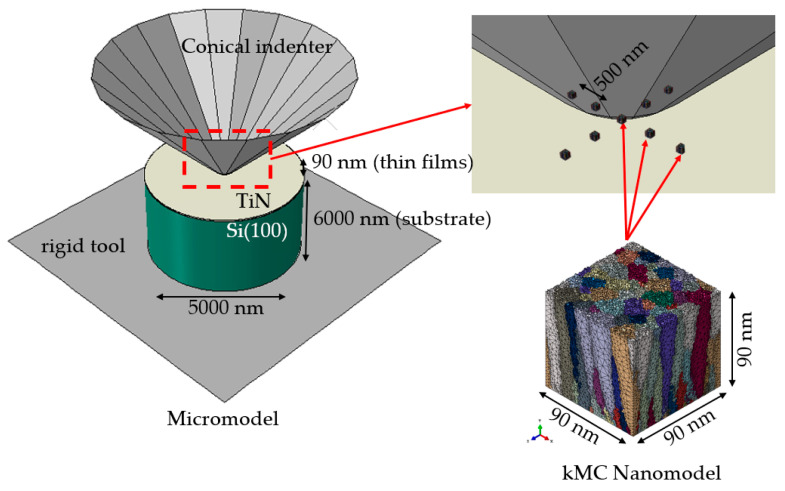
Schematic illustration of the multiscale model of the nanoindentation test.

**Figure 10 materials-14-01705-f010:**
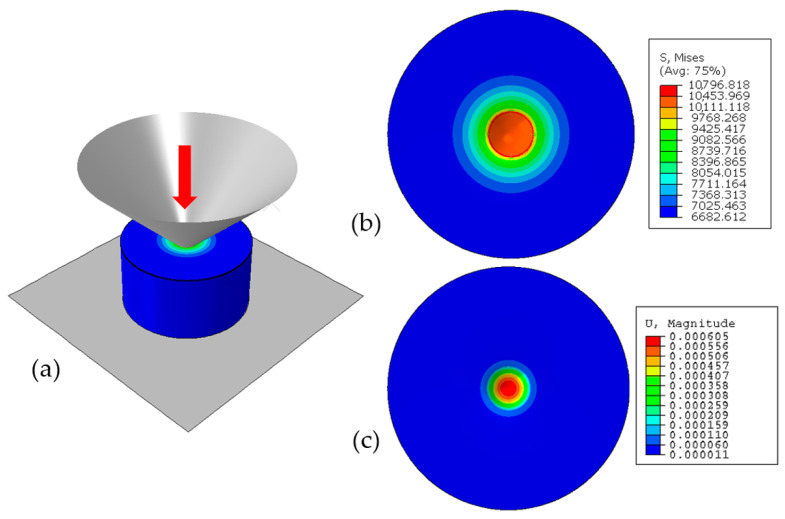
Distribution of (**a**) equivalent stress (MPa) in the model from the side view and, (**b**) equivalent stress (MPa), (**c**) displacement (mm) from the top view.

**Figure 11 materials-14-01705-f011:**
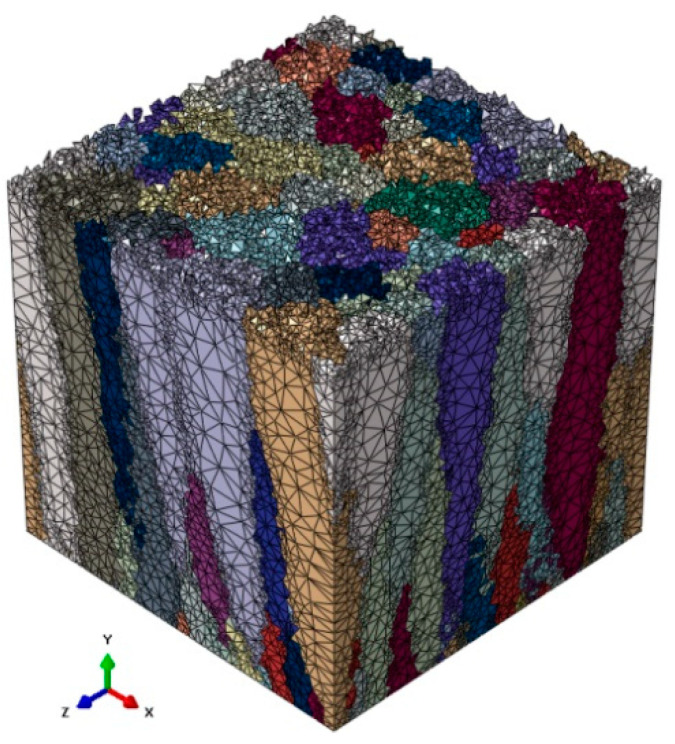
Finite element mesh of the investigated TiN thin film columnar morphology.

**Figure 12 materials-14-01705-f012:**
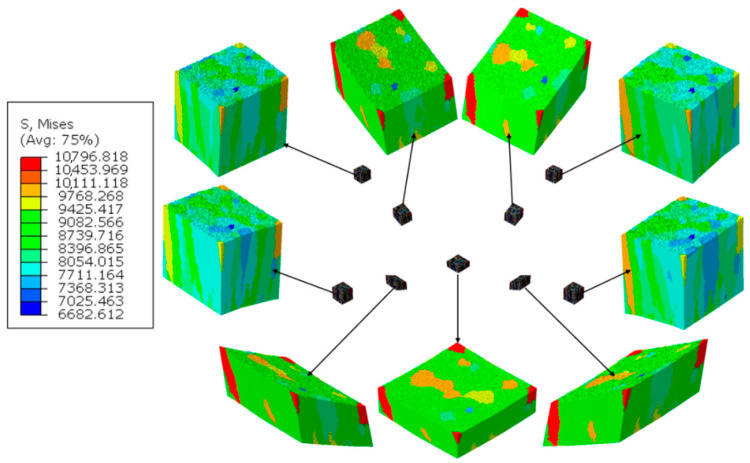
Equivalent stress distribution within the DMR (Digital Material Representation) model after nanoindentation test.

**Figure 13 materials-14-01705-f013:**
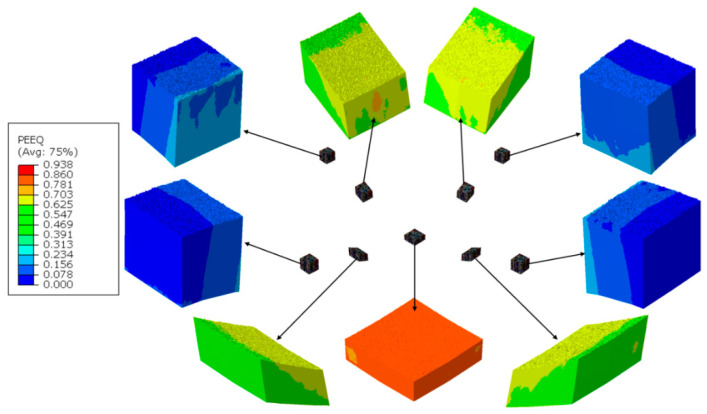
Equivalent strain distribution in the DMR model after nanoindentation test.

**Figure 14 materials-14-01705-f014:**
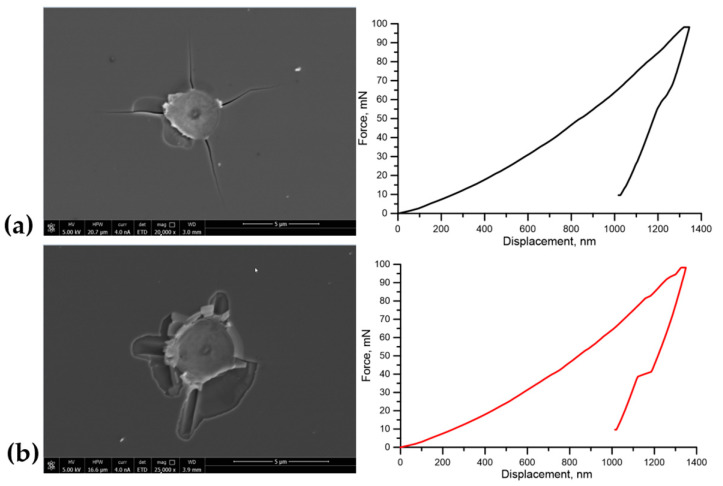
SEM images (Quanta 3D 200i, FEI, Hillsboro, OR, USA) revealing fractures in the (**a**) Si substrate and (**b**) TiN thin film after nanoindentation test with force–displacement curves received during test.

**Table 1 materials-14-01705-t001:** Parameters used in the kMC PLD model of the TiN/Si deposition.

Parameter	Value
Domain edge length	90 nm
Elementary cell size	1 nm
Substrate melting temperature	1414 °C
Substrate temperature	200 °C
Binding energy	0.8 eV
Deposition rate	0.05 nm/s
Vibration frequency	1 × 10^13^ Hz

**Table 2 materials-14-01705-t002:** Measured columns’ dimensions in the TiN thin film (width, height).

Dimension Number	1	2	3	4	5	6	7	8	9	10	11	12
Size (nm)	15.1	25.9	89.1	24.8	33.1	94.4	23.6	30.5	87.6	30.8	53	93.6

## Data Availability

The raw/processed data required to reproduce these findings cannot be shared at this time as the data also forms part of an ongoing study.

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
