# Peer review of "Numerical Study on the Dependency of Microstructure Morphologies of Pulsed Laser Deposited TiN Thin Films and the Strain Heterogeneities during Mechanical Testing"

_materials, 2021, doi:10.3390/ma14071705_

Round 1

Reviewer 1 Report

the paper is presenting an interesting topic on relation between the morphology and strain level in TiN thin film materials . The few comments are due before we proceed with this manuscript:

-the title is misleading: i suggest:

Numerical study on the dependency of microstructure morphologies of pulsed laser deposited TiN thin films and the strain heterogeneities during mechanical testing.

-the abstract is misleading. have you used FEM or monte-Carlo or how did u mix both. this must be clearly explained in abstract.

-little was said about te region of interest as this is hugely important and influential in such numerical analysis. please describe how many areas were tested in order to choose this presented one in the paper.

-in fig. 14, whats the force, time and displacement values?

-about the nanoindentation, citation is required to 

  • DOI: 10.1016/j.promfg.2020.04.127

since the large displacement In the TiN film, especially between columns, it would be easy to induce large strain and stress irregularities. I am not sure Prediction of these regions is of a novelty in this work as it will have a high probability anyway. please explain.

when authors take my comments in and answer my questions, i can reconsider in my decision.

Author Response

Dear Reviewer,

We would like to sincerely express our gratitude for the detailed review and valuable comments. As a result, we feel that the manuscript is improved and all shortcomings are eliminated. All responses to subsequent comments are presented in the attachment , while changes made to the manuscript are marked in red in the corrected text.

Best regards,

Reviewer 2 Report

The authors reported a development of a numerical model of digital material representation of thin film microstructure morphology of the ALD deposition process. Since the commonly used numerical models of deformation today, neglect the inner structure of deposited thin films, which depreciates the quality of obtained data. For example, the deposited material under loading conditions is usually defined as isotropic without taking into account columns and surface wrinklings. Thus, simplified models cannot give sufficient information about material resistance to deformation. Therefore, the development of the model, which precisely maps thin films' morphologies and inner structures during modelling of exploitation conditions, seems extremely important to obtain results comparable with those from experimental investigations.

This manuscript is an extension of authors’ last publication in Thin Solid Films (2019). Therefore, this study should be more elaborated into the calculation method, so that it can be suggested to be published in this journal. See followings:

  1. The nano-columnar structure generated by kMC simulation with different crystalline facets and elastic constant and Poisson ratio of each columnar structure should be different due to the different crystalline orientation of each column. In the manuscript, this part did not mentioned clearly. How the elastic modules and Poisson ratios defined at different columns and at the boundaries of columns.  
  2. The acronym of DMR(digital material representation) is not defined at first sight in the text. Also the FE(finite element) is not defined either.

Line 272: The name of the code Abaqus should be ABAQUS.

Author Response

Dear Reviewer,

We would like to sincerely express our gratitude for the detailed review and valuable comments. As a result, we feel that the manuscript is improved and all shortcomings are eliminated. All responses to subsequent comments are presented in the attachment, while changes made to the manuscript are marked in red in the corrected text.

Best regards,

Round 2

Reviewer 1 Report

my comments have been been inserted and my questions were answered so I can now accept the paper for publication